

**Comparison of CSES ionospheric RO data with COSMIC measurements**
Xiuying Wang[1], Wanli Cheng[2], Zihan Zhou[1], Song Xu[1], Dehe Yang[1], Jing Cui[1]
1. Institute of Crustal Dynamics, China Earthquake Administration, Beijing, China
2. Xinyang Station, Henan Earthquake Administration, Henan, China
Corresponding author: Xiuying Wang
**Abstract**: CSES is a newly launched electric-magnetic satellite in China; its main scientific objective
is to monitor earthquake related disturbances in ionosphere. A GNSS occultation receiver (GOR) is
installed on the satellite to inverse electron density related parameters. In order to validate the
radio occultation (RO) data from GOR onboard CSES, a comparison between CSES RO and the co-
located COSMIC RO data is conducted to check the consistency and reliability of the CSES RO data
using measurements from February 12, 2018 to March 31, 2019. CSES RO peak values ($N_mF_2$), peak
heights ($h_mF_2$), and electron density profiles (EPDs) are compared with corresponding COSMIC
measurements in this study. The results show that: (1) $N_mF_2$s between CSES and COSMIC are in
extremely good agreement with a correlation coefficient of 0.9891. The near zero bias between
the two sets is $0.01235 \times 10^5/cm^3$ with a RMSE of $0.3680 \times 10^5/cm^3$; and the relative bias is 2.14%
with a relative RMSE of 16.40%, which are in accordance with previous studies according to error
propagation rules. (2) $h_mF_2$s between the two missions are also in very good agreement with a
correlation coefficient of 0.9379; the mean difference between the two sets is 0.73km with a RMSE
of 13.02 km, which is within the error limits of previous studies; (3) Co-located EDPs between the
two sets are generally in good agreements, but with a better agreement for data above 200km
than that below this altitude. Data at the peak height ranges show the best agreement, and then
data above the peak regions; data below the peak regions, especially at the altitude of about the
E layer, show relatively large fluctuations. It is concluded that CSES RO data are in good agreement
with COSMIC measurements, and the CSES RO data are applicable for most ionospheric-related
studies. However, particular attention should be paid to EDP data below peak regions in application.
**Key words**: CSES satellite; COSMIC mission; radio occultation; validation; ionosphere
**1. Introduction**
The first China Seismo-Electromagnetic Satellite (CSES), also called ZH-1 in China, has been
working for over 1 year since its launch on February 2, 2018. This satellite is the first spaced-based
platform in China for both the 3-D earthquake observation and geophysical field measurement; a
subsequent satellite of this series will be launched in 2022 and the engineering work is under way.
The primary scientific objectives of the CSES mission is to obtain world-wide data of space
environment of the electromagnetic field, ionospheric plasma and charged particles, to monitor
and study the ionospheric perturbations which may possibly associated with earthquake activity,
especially with those destructive ones, to support the research on geophysics, space sciences as
well as electric wave sciences and so on, and also to provide the data sharing service for
international cooperation and scientific community (Shen et al., 2018; Wang et al., 2019).
The CSES satellite is sun synchronous orbit with an inclination angle of 97.4° at the altitude
of 507 km. The local time of descending and ascending nodes are 1400 and 0200 respectively. It
takes about 94.6 minutes to complete a circular orbit, thus about 15 orbits per day. The revisiting



period of CSES is 5 days, which means the satellite will nearly repeat the orbits after 5 days. At present, the observation range of the CSES satellite is mainly between -65° and +65° of geographic latitudes.

There are eight Chinese payloads and one Italian payload onboard the CSES satellite, belonging to 3 categories: (1) electromagnetic observations, including search-coil magnetometer (SCM), electric field detector (EFD), and high precision magnetometer (HPM); (2) ionosphere related observations, including GNSS occultation receiver (GOR), plasma analyzer package (PAP), Langmuir probe (LAP), and tri-band beacon (TBB); (3) and high-energy particles observations, including high energetic particle package (HEPP) and high energetic particle package detector (HEPD), of which HEPD is provided by Italian Space Agency.

Of the eight payloads, four are related to ionospheric parameter observations. The GOR payload onboard CSES is a GPS/BD2 receiver to inverse ionospheric electron densities according to the radio wave refractivity when traversing the ionosphere. It is known that low Earth Orbit (LEO) based GPS/GNSS radio occultation (RO) technique has been a powerful technique in ionosphere monitoring; using this technique, the accurate electron density profiles (EDPs) in the ionosphere can be derived with high vertical resolution on a global scale from bending information of the RO signals (Kuo et al., 2004; Rocken et al., 2000; Schreiner et al., 1999). Therefore, many LEO satellites were launched with RO payload after the pioneer RO experiment on GPS/MET mission (Hajj et al., 1998; Schreiner et al., 1999), such as the CHAMP satellite (Jakowski et al., 2002; Wickert et al., 2009), the GRACE satellites (Beyerle et al., 2005), the most famous COSMIC mission (Anthes et al., 2008; Lei et al., 2007), and so on. The application of RO technique is also an important part of the CSES satellite. Combining with the in situ electron density measurements onboard CSES, the CSES RO retrieved electron densities can be used to study global scale ionospheric 3D images from the bottom of the ionosphere to the altitude of the CSES satellite using the large amount of daily occultation events. However, a complete and thorough validation of the RO measurements obtained by the CSES satellite is a necessary work before the retrieved electron density profiles can be used for ionospheric studies.

A primary comparison, between CSES and COSMIC using the global distribution of peak values ($N_mF_2$) and peak height ($h_mF_2$) data, was carried out during the in-orbiting test period of the CSES satellite, and the CSES $N_mF_2$ values were also compared with the measurements from 3 digisondes in China (Cheng et al., 2018). Both the comparisons show that the CSES RO $N_mF_2$ data are generally consistent with data from other measurements. However, as the comparisons are limited to the peak values and the date coverage is only two months, a more complete validation is still required to assess the consistency and reliability of the RO profiles obtained by the CSES satellite. A large amount of RO profiles have been obtained so far by CSES, which provide enough data to implement a more detailed validation work.

Validation of RO profiles is usually done by comparing the profiles with the measurements from vertical ionosondes or incoherent scatter radars (ISRs). However, RO electron density profiles above the F2 peak region cannot be validated by ionosonde observations due to the unreliable extrapolating data at these altitudes. In addition, the uneven distribution of the ionsonde stations, most located on inland and fewer in the oceans, restricts the global comparison work. Although ISRs can be used to validate RO electron density profiles above F2 peak region, this comparison is limited due to the relatively small number of ISR sites as well as their limited operating time. Therefore, we will carry out the comparison work using the RO measurements from the COSMIC





dataset in this paper.
Validation of the COSMIC electron density measurements has been performed in numerous
studies using different measurements, such as the cross validation of the retrieved profiles from
nearby spacecraft in the same COSMIC mission (Schreiner et al., 2007), comparison with ground-
based ionosondes and ISRs (Cherniak and Zakharenkova, 2014; Chu et al., 2010; Chuo et al., 2011;
Habarulema et al., 2014; Kelley et al., 2009; Krankowski et al., 2011; Lei et al., 2007; McNamara
and Thompson, 2015), comparison with the in situ electron density measurements (Lai et al., 2013;
Pedatella et al., 2015; Yue et al., 2011), comparison with radio tomography data using space
climatology phenomenon (Thampi et al., 2011), comparison with ionospheric model IRI (Lei et al.,
2007; Wu et al., 2015; Yang et al., 2009), and so on. As COSMIC RO data have been extensively
validated and widely accepted for application, COSMIC RO data are used to validate the in situ
plasma density observations from the Swarm constellation (Lomidze et al., 2017). We therefore
also try to use the COSMIC RO dataset to validate CSES RO measurements because of its relative
large amount of data with globally spatial coverage. In addition, similar RO retrieved data from the
two sets also provides a unique opportunity to check the consistency and reliability of CSES profiles
except for $N_mF_2$ and $h_mF_2$ parameters.
In this study, the validation work is implemented by comparing CSES $N_mF_2$, $h_mF_2$, and data from
EDPs at some altitudes with corresponding COSMIC measurements, and the bias and RMSE
between the two sets are then calculated and estimated to evaluate the consistency and reliability
of CSES RO inversed data. Based on the results, an application suggestion is given on the CSES RO
data.
**2.  Data and Method**
**2.1 CSES and COSMIC RO data**
**1. CSES RO data**
GOR payload onboard CSES can receive the dual frequencies from GPS (L1:
1575.42MHz±10MHz; L2: 1227.6MHz±10MHz) and DB2 (L1:1561.98MHz±2MHz; L2:
1207.14MHz±2MHz) to inverse atmospheric and ionospheric parameters with sampling rate of
100Hz and 20Hz respectively. Firstly, TECs from GPS to LEO are calculated from the carrier phase of
the dual frequencies; and then electron densities are retrieved from TECs using the Abel integration
transformation. The Abel integration method and assumptions used in RO inversion process have
been described in detail in many publications (Kuo et al., 2004; Lei et al., 2007; Schreiner et al.,
1999) and will therefore not repeat here.
The GOR payload onboard CSES started to work on February 12, 2018 and ionospheric radio
occultation (RO) measurements have been conducted since then. CSES RO inversed data are
divided into 5 levels: 0, 1, 2, 2A and 3. Level-0 is original data; Level-1 is physical quantity in time
order; Level-2 is physical quantity data with satellite orbital information and geomagnetic
coordinates, while Level-2A is similar with Level 2, but with higher precise orbital information; and
Level-3 is 2D structural data product from Level-2 and Level-2A, which can provide peak value, peak
height and EDP data.



All the CSES RO data of the 5 levels are saved in HDF5 format, which is organized in a
hierarchical way. One file is saved for each occultation event, and about 500 to 600 occultation
event files can be obtained per day. Data users can refer to the data specification document for
detailed description of data file naming conventions and data level classification, which can be
obtained from the CSES data sharing center website www.leos.ac.cn.
More than 180,000 CSES occultation profiles have been obtained from 2018-02-12 to 2019-
03-31, of which occultation events co-located with that from the COSMIC mission will be used to
carry out the comparison and validation work in this paper.
**2. COSMIC RO data**
The COSMIC (Constellation Observing System for Meteorology, Ionosphere, and Climate, also
called FORMOSAT-3 in Taiwan) mission, a constellation of six identical low Earth orbit satellites
launched in April 2006, is a joint Taiwan-US mission to observe the near-real-time GPS RO data
(Anthes et al., 2008). COSMIC RO data come from the GPS Occultation Experiment (GOX) receivers
onboard the COSMIC satellites that monitor the two GPS L-band signals to establish the relative
geometries of satellite positions and differences in phase/Doppler shifts (Rocken et al., 2000). At
the University Corporation for Atmospheric Research (UCAR) COSMIC Data Analysis and Archive
Center (CDAAC), ionospheric profiles are retrieved by use of the Abel inversion technique from TEC
along LEO–GPS rays. Detailed description of CDAAC data processing and EDP retrieval method can
be found in some literatures (Kuo et al., 2004; Lei et al., 2007).
In the present study, the COSMIC level-2 electron density profiles provided as "ionPrf" files
from 2018-02-12 to 2019-03-31 are used, which can be downloaded from CDAAC website
http://cdaa-www.cosmic.ucar.edu/. COSMIC can provided over 2000-2500 RO profiles per day at
its initial stage, but for now only 200-300 events on average can be obtained each day. Fig.1 gives
the total occultation numbers of each month for both CSES and COSMIC missions from February
2018 to March 2019.

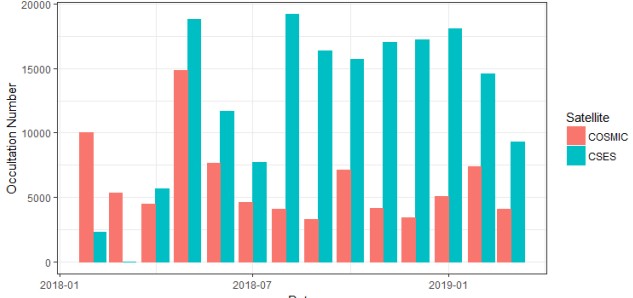


Fig. 1 Occultation number per month from February 2018 to March 2019 for both CSES and COSMIC
From Fig. 1 it can be seen that over 15,000 occultation events can be obtained by CSES each
month, or over 500 per day on average, after the initial in-orbit testing stage from February 2018
to July 2018. In contrast, occultation numbers from COSMIC are much less, there are only about
200 occultations on average each day. A total of over 86,000 occultation events have been obtained
from the COSMIC data center from February 2018 to March 2019.
Based on these two datasets from CSES and COSMIC, the co-located occultations within
defined spatial and temporal criteria from the two measurements are selected and used to carry
out the comparison work.



**2.2 Data selection**

2       In order to make the comparison between CSES and COSMIC RO data as accurate as possible,
spatial and temporal criteria must be defined to select matching occultation profiles for
subsequent comparison analysis.
5       Before determining the selection criteria, it should be pointed out here that RO retrieved
electron density profiles cannot be interpreted as actual vertical profiles because both the LEO and
GPS are in motion during the occultation process. The geographic location of the tangent points of
a profile may vary in several hundred kilometers, which means the spatial range of a profile can
cover several degrees in horizontal latitude and longitude range, and several hundred kilometers
in vertical altitude range. However, the ionospheric spatial correlation can extent to a large area as
suggested by some researches (Shim et al., 2008; Yue et al., 2007). According to Shim et al. (2008),
the daytime meridional correlation lengths are approximately 9° and 5° at mid- and low-
latitudes, and the nighttime values are about 3° and 2° at mid- and low latitudes, respectively;
the zonal correlation lengths are 23° at mid-latitudes and 15° at low latitudes during the day,
and are 11° at mid-latitudes and 10° at low latitudes during the night. Therefore, the matching
profile pairs from the two missions must be within the correlation distances. Considering the
relatively small number of occultation events from the COSMIC measurements, we define the
search criteria for co-located occultation events as follows: (1) the time difference between the
matching occultation pairs is less than 30 min; (2) the distance differences between the locations
of the two occultation events are within $2° \times 6°$ range in latitudinal and longitudinal directions.
Here, the tangent point at F2 peak value of an occultation profile is defined as the location of the
occultation event. The reason to use the peak value tangent point as the occultation location is
because the peak value is normally located at the middle of a profile for the CSES EDPs, and by this
way the spatial differences of the corresponding points, especially the top and bottom points,
between the matching profile pairs can be limited to the correlation distance range as many as
possible.
Based on the above criteria, the RO profiles from CSES and COSMIC, covering the period from
February 2018 to March 2019, are searched to select the co-located profile pairs. The profiles with
$N_mF_2$ appearing below 150km or above 500 km are discarded, and profiles with only ascending or
descending part of a profile which cannot determine the peak values are also deleted from the
CSES dataset. A total of 891 matching profiles are found, and their distributions are given in Fig. 2.
Numbers of occultation in each 10 latitudinal region are also calculated and given in Fig.3.

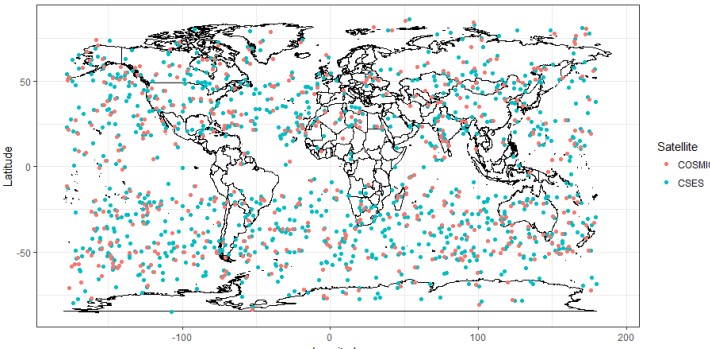






Fig. 2 Distribution of the selected profile pairs
(Each dot indicates the location of the tangent point of the maximum values in a profile.)

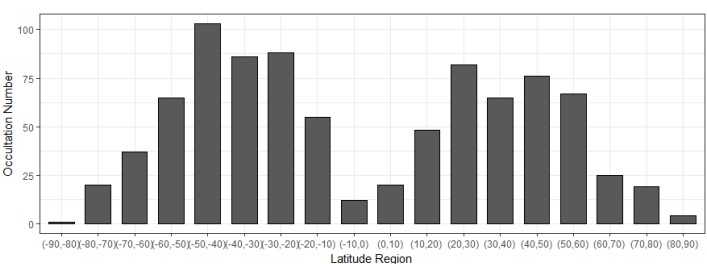

Fig. 3 Number of co-located profile pairs along latitudinal regions

5        From Fig.2, it can be seen that the selected profile pairs are globally distributed, which makes

the data be representative of the whole dataset on spatial scale. In addition, the over a year
temporal segment, covering different periodic components of the ionospheric variations, makes
the data involved in the comparison be temporal representative also.

9        It is necessary to note that because the CSES satellite is sun-synchronous orbit as mentioned

earlier, the local time of the occultation events is concentrated around the ascending (0200) and
descending (1400) local time. Special attentions should be paid on the local time issue when CSES
and COSMIC RO data are combined together.

13       Another point to note is that most of the selected profile pairs are distributed in the mid-

latitude regions, as shown in Fig. 2 and Fig. 3, and the equatorial region as well as the high latitude
regions exhibit lower number of occultation events, which ensures that the selection criteria can
be satisfied for most of the selected matching profiles.
**2.3 Comparison method**

18       The CSES RO electron density data are compared with the co-located COSMIC RO data to assess

the consistency and reliability of the CSES RO data relative to that of the COSMIC, and then the
consistency and reliability of the CSES RO data relative to ground-based measurements are
estimated using the results obtained by previous researches on COSMIC RO data according to error
propagation rules.

23       The maximum electron density and its height, namely $N_mF_2$ and $h_mF_2$ from CSES RO data, are

compared and analyzed directly with the corresponding co-located COSMIC data, respectively.
Besides RO peak values, the profiles of the matching pairs are also compared in this study. To
compare the similarities of the profiles, average electron density data near some special altitudes
of a profile are calculated and compared. Because the orbit altitude of CSES is 507, only data below
this altitude are obtained from the CSES RO retrieved EDPs. Therefore, the special altitudes
involved in the comparison include 100, 150, 200, 250, 300, 350, 400, 450 and 500 km. The
consistency and reliability of the CSES RO profiles are thus evaluated by combining the comparison
results of these special altitudes.

32       Normally, the height resolution in the F region has the order of 20 km for the COSMIC RO (Kuo

et al., 2004), but CSES RO data has a higher resolution due to the higher sampling rate of the radio
signals. We therefore use the average data between the selected altitudes±10km, which is just
within the vertical resolution of the COSMIC RO data.





1       In this study, all the selected matching profiles are involved in the analysis rather than those
2       observed in geomagnetic quiet days. In this way, disturbed data caused by events such as
3       geomagnetic storms can also be used to compare their similarities/differences under these special
4       occasions.

**3.    Results and Discussions**
**3.1 Comparison of $N_mF_2$**

7       The maximum electron density in the ionospheric F2 layer, $N_mF_2$, is the most important
8       parameter in ionospheric related studies. To compare this parameter, the maximum electron
9       density data are extracted from all the matching RO files of CSES and COMSIC measurements.
Scatter plot of these matched $N_mF_2$ points is given in Fig. 4, also given is the histogram of the data
differences between the matched peak value points. As shown in Fig. 4b, data differences between
the two measurements are normally distributed; points with data differences exceeding 3 times
root mean square error (RMSE), shown as hollow circles in Fig. 4a, are considered outliers and can
be eliminated from the selected dataset according to $3\sigma$ rule. Red points in Fig. 4a are peak values
observed during geomagnetic storm conditions of Dst<-30 nT, all of which are within $3\sigma$ limits and
matched very good as shown in Fig. 4a. Fig. 4a also gives the linear fitting equation, the goodness-
of-fit coefficient $R^2$ (square of correlation coefficient), and number of data points.

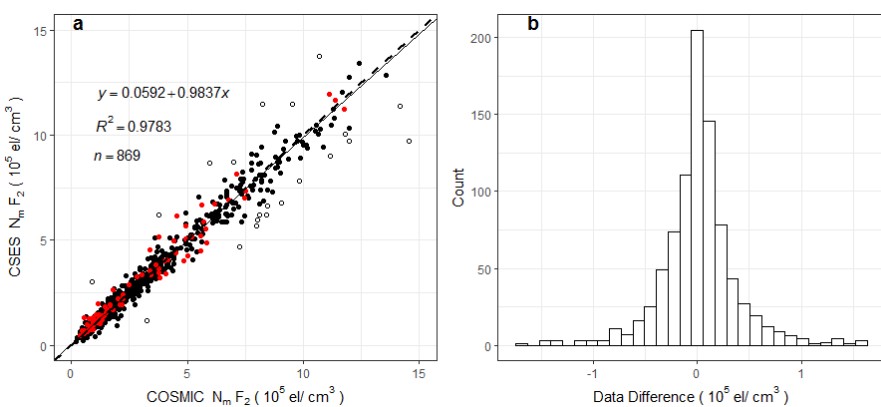

19          Fig. 4 Scatter plot of matched $N_mF_2$s and histogram of the data differences between the two sets

(The dash line in Fig. 4a is the equal value line with a slope of 1, and the solid line is the linear fitting line. Hollow
circles are points exceeding 3 times RMSE. Red solid points are data observed when Dst<-30nT. y refers to CSES
$N_mF_2$ data, x COMSIC $N_mF_2$ data. $R^2$ is the goodness-of-fit coefficient; n is the total data number.)

23          The correlation coefficient between the matched $N_mF_2$s with elimination of outliers is 0.9891,

and correlation coefficient without elimination of outliers is 0.9786, both of which can pass the
significance test of confidence level 0.01. The high correlation coefficient indicates the high
consistency between the two $N_mF_2$ sets. The linear fitting coefficient of 0.9837 given in Fig. 4a is
very close to 1; the data differences between the two sets are nearly normal distributed as shown





in Fig. 4b, and most of the data differences is around zero, all of which means that the CSES $N_mF_2$s
are nearly equal to COSMIC $N_mF_2$s with a nearly zero bias. Both the correlation coefficient and the
liner fitting coefficient indicate that the CSES $N_mF_2$s are in extremely good agreement with the
corresponding COSMIC data.
To quantify the error, we also calculate the RMSE and relative RMSE between the two sets. The
mean of the data differences between CSES $N_mF_2$ and COSMIC $N_mF_2$ is $0.01235 \times 10^5/cm^3$, and the
RMSE between the two matched datasets is $0.3680 \times 10^5/cm^3$, both of which are very small when
comparing with the original data. Therefore, the nearly zero bias between the two measurements
of $N_mF_2$ can be neglected, which is in accord with the normal distribution with most data
differences clustering around zero as shown in Fig. 4b. The mean relative differences of $N_mF_2$
calculated from equation (1) is 2.135%, and the corresponding relative RMSE is 16.40%. The mean
relative data differences is also extremely small. The mean of data differences and the mean of
relative data differences, as well as their RMSEs, again show that the CSES RO data are in very good
agreement with the COSMIC data.

$$E_r = \frac{1}{n} \sum_{i=1}^{n} \frac{z_i - y_i}{y_i} \times 100\% \quad (1)$$

Where $z_i$ refers to the $i$th CSES $N_mF_2$ data of the matched data pairs, and $y_i$ the corresponding
COSMIC $N_mF_2$ data.
To compare the difference of correlation relationship for daytime and nighttime data, the data
in Fig. 4 are divided into two groups. As introduced in section 2.2, the local time of CSES satellite is
fixed at 0200 during night and 1400 during day, and the local time of RO data are around these two
fixed local time, we therefore don't need to further consider differences caused by different local
time.
The scatter plots for daytime and nighttime data are drawn using the same method introduced
above and given in Fig. 5. The data obtained under geomagnetic storm conditions are also shown
in red color, all of which are within the 3σ limits.

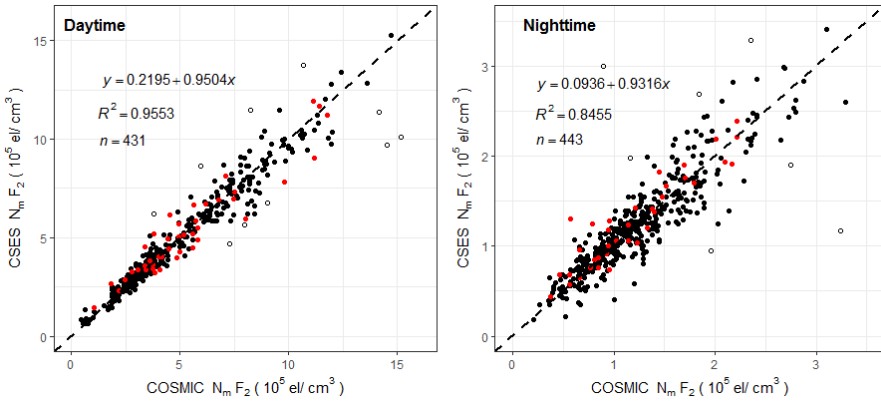


27                      Fig. 5 Scatter plot of $N_mF_2$ for daytime and nighttime data

(the dash line in Fig. 5 is the equal value line with a slope of 1)
Correlation coefficient for daytime data with elimination of outliers is 0.9774, and 0.9637
without elimination of outliers; for nighttime data with elimination of outliers, correlation





coefficient is 0.9195, and 0.8767 for all the data. The higher daytime correlation coefficient indicates a better agreement for the daytime data than the nighttime data. This can be seen clearly from Fig.5, the nighttime data are obviously fluctuated more violently.

The mean data differences for daytime data is $-0.01634 \times 10^5$ /cm³ with a RMSE of $0.5572 \times 10^5$ /cm³ , and mean data differences for nighttime data is $0.01010 \times 10^5$ /cm³ with a RMSE of $0.2094 \times 10^5$ /cm³. The opposite sign of the daytime and nighttime mean data differences indicates that the CSES daytime data is slightly smaller than that of the COSMIC, while CSES nighttime data is slightly greater than the corresponding COSMIC data, but both the means of data differences are extremely small and can be consider zero bias when comparing when the original measurements.

Table 1 Absolute and relative error of $N_mF_2$ between CSES and COSMIC

| | Correlation coefficient | Absolute Error | | Relative Error | |
|---|---|---|---|---|---|
| | | Mean (/cm³) | RMSE (/cm³) | Mean | RMSE |
| Total | 0.9891 | $0.01235 \times 10^5$ | $0.3680 \times 10^5$ | 2.135% | 16.40% |
| Daytime | 0.9774 | $-0.01634 \times 10^5$ | $0.5572 \times 10^5$ | 1.344% | 12.85% |
| Nighttime | 0.9195 | $0.01010 \times 10^5$ | $0.2094 \times 10^5$ | 2.492% | 18.70% |

When comparing the different results given in Table 1, the absolute mean data differences for daytime data is slightly greater than that of the overall result, and with an obvious larger RMSE; while the mean data differences for nighttime data is slightly smaller than the overall result, and also with a smaller RMSE. However, the two plots in Fig.5 indicate that the daytime data is obvious better than the nighttime data. This is because the daytime data are much greater than nighttime data, absolute error cannot correctly reflect the real situation when comparing data values with different magnitudes. We therefore calculate the relative errors for both the daytime and nighttime data. The mean relative data differences for daytime data is 1.344% with a relative RMSE of 12.85%, and mean relative data difference for nighttime data 2.429% with a relative RMSE of 18.70%, which indicate an obvious better agreement for the daytime measurements.

It is necessary to point out that most of the daytime data points with higher values are located below the dash line as shown in Fig. 5, which means that the COSMIC $N_mF_2$s are larger than that of the CSES, so there is a negative bias between the two sets; while for nighttime data, most the data points with higher values are above the dash line, indicating greater CSES $N_mF_2$ values, thus there is a positive bias between them. This can also explain why there is a higher correlation coefficient when combining daytime and nighttime data together.

There is another point to point out. As can be seen from Table 1, the absolute mean difference for daytime data is negative, while the mean relative differences is positive. Further analysis shows this different signs is caused by some points with much larger CSES $N_mF_2$ values.

Here, we compare our results with previous studies.

Lei et al. (2007) obtained a correlation coefficient of 0.85 when comparing COSMIC $N_mF_2$ with observations from 31 globally distributed SPIDR (The Space Physics Interactive Data Resource, http://spidr.ngdc.noaa.gov/spidr) ionosondes using data observed in July 2006. Chuo et al. (2013) demonstrated that COSMIC derived $N_mF_2$ values are in good agreement with digisonde observations of different seasons; they also reported an agreement about 0.96 using observations from a lower latitude ionosonde in south hemisphere using a big dataset from May 2006 to April 2008. Chu et al. (2010) found a correlation coefficient of 0.98 when comparing $N_mF_2$s between COSMIC and 60 globally distributed ionosondes belonging to SWPC (Space Weather Prediction Center), NOAA using data from November 2006 to February 2007. Krankowski et al. (2011)





obtained a very good correlation coefficient of 0.986 when validating COSMIC RO data in 2008
using measurements in European mid-latitude ionsondes. Our result of 0.9891 is quite similar to,
or even slightly better than those results, when considering the similar solar activity levels. A
relative high correlation coefficient between CSES $N_mF_2$ and ionosondes can be deduced since the
correlation transitive conditions are satisfied according to Langford et al. (2001). We therefore
obtained that CSES RO derived peak values are in very good agreement with COSMIC and ground-
based measurements.

8         For $N_mF_2$ relative errors, Krankowski et al. (2011) obtained a mean relative bias of 0.72% with a
standard deviation of 8.42%, and the slope of the linear fitting line is 0.994 using a manual
screening dataset in Europe, which is better than the results in this paper. Wu et al. (2009) got a -
3.2% relative bias with a standard deviation of 20.7% when comparing $N_mF_2$s between COSMIC and
62 global ionosondes from SPIDR using data from July 2006 to Decemeber 2007. Yue et al. (2011,
2013) suggest that the ability to retrieve $N_mF_2$ using the Abel inversion technique has an
uncertainty about 10%. Based on the linear fitting equation between CSES and COSMIC and error
propagation rules, we can deduce that the relative errors between CSES peak values and ground-
based measurements are comparable to prior studies.

17        As to the absolute error, Kelley et al. (2009) obtained a RMSE of $1.0 \times 10^5$ /cm$^3$ when comparing
COSMIC data with ISR; Hajj et al. (2000) obtained a $N_mF_2$ RMS difference of about $1.5 \times 10^5$ /cm$^3$
when comparing the GPS/MET measurements with nearby ionosonde data, and Jakowski et al.
(2002) also obtained a similar RMS difference of about $0.9 \times 10^5$ /cm$^3$ when comparing the CHAMP
RO measurements to the in situ Langmuir probe data on the same satellite. Habarulema et al. (2014)
suggested that all RO data sets are close to the ionosonde data within similar error margin for both
mid-latitude and low-latitude regions when comparing COSMIC, GRACE and CHAMP RO data with
that of ionsondes. The absolute errors of our results are much smaller than these results, indicating
an extremely good agreements between CSES and COSMIC RO $N_mF_2$ and further confirming that
CSES RO are also within the general error limit as proposed by Habarulema et al. (2014).

27        Better result of daytime data in this study is in accord with the conclusion obtained by Wu et
al. (2009) and Yue et al. (2011). As we know, the nighttime data has a more complex spatial
distribution pattern compare to daytime data although daytime data are affected by solar radiation
during day time. Larger inversion error will be produced when facing uneven spatial distribution of
electron density due to the spherical symmetry assumption of the Abel inversion method. The
complex night time spatial distribution can also be proved by the smaller correlation distance
during nighttime than that of daytime as discussed in section 3.2 (Shim et al., 2008).

34        Besides data obtained under geomagnetic quiet days, data obtained under geomagnetic storm
conditions are also quite consistent with each other, this conclusion is also supported by the results
from Hu et al. (2014). They suggested that COSMIC measurements are acceptable under
geomagnetic disturbed conditions when comparing COSMIC RO data with observations from Sanya,
a lower latitude ionosonde in China.

39        As suggested by Schreiner et al. (2007) that co-located RO soundings allow the precision of the
technique to be estimated, but not the accuracy. That fact that the nearly zero bias for both
daytime and nighttime data and for the overall data, the normal distribution of the data differences,
as well as the extremely high correlation coefficient between CSES $N_mF_2$ and COSMIC $N_mF_2$,
demonstrates that the CSES $N_mF_2$ data are highly consistent and identical with COSMIC
measurements, even under geomagnetic storm conditions, indicating a similar precision of CSES





RO $N_mF_2$ data as that of COSMIC. Given the reliability (accuracy) of the COSMIC data proved by
many studies, we believe that the CSES $N_mF_2$ measurements are also quite reliable. Since the co-
located data points are globally distributed, the comparison results can be generalized to the
overall CSES $N_mF_2$ dataset obtained so far.
**3.2 Comparison of $h_mF_2$**

6         The height of the maximum peak values in F2 layer, $h_mF_2$, is also a very important parameter
for ionospheric studies. We therefore also compare this parameter using the corresponding
COSMIC dataset.
9         Comparison of the $h_mF_2$ values between the two sets using the same method as that by $N_mF_2$,
the scatter plot of $h_mF_2$ and the histogram of the data differences are given in Fig. 6. Data points
exceeding 3 times of RMSE, shown as hollow circles in Fig. 6a, can be deleted from the selected
data sets when calculation is implemented. Again, all the peak height points obtained under
geomagnetic condition (red points) are within the 3 $\sigma$ limits as shown in Fig. 6a. It can be seen
clearly From Fig. 6a that most of the outliers (hollow circles) are obviously above the dash line,
which means that occasionally RO data from the CSES dataset will much overestimate $h_mF_2$ values.

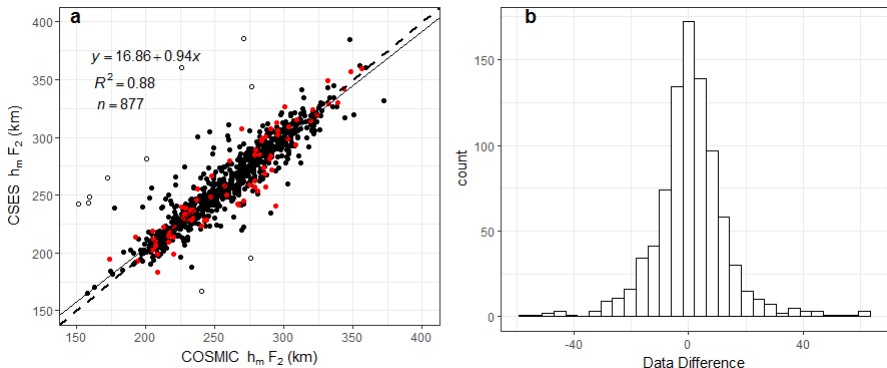

Fig. 6 Scatter plot of $h_mF_2$s for CSES and COSMIC and histogram of their differences
(The dash line is the equal values line with a slope of 1, and the solid line is the linear fitting line. y refers to the
CSES $h_mF_2$, and x COSMIC $h_mF_2$. Hollow circles are points exceeding 3 times standard deviation of data differences
between matched points. Red points are peak height obtained under geomagnetic condition of Dst<-30 nT. )
21        The correlation coefficient of $h_mF_2$ is 0.9379, though slightly lower than that of the $N_mF_2$, but
can also pass the significance test of confidence level 0.01, which also indicates a very good
agreement between the two sets of $h_mF_2$. The mean of the $h_mF_2$ data differences (CSES $h_mF_2$ minus
COSMIC $h_mF_2$) is 0.73 km, which indicates a slight greater $h_mF_2$ for the CSES peak height values; and
the RMSE is 13.02 km. $h_mF_2$ data difference between the two sets is so small, which can be regarded
as nearly zero bias.
27        Compared with $N_mF_2$, $h_mF_2$ data fluctuate more violently. It can be seen from Fig. 6a that some
data points are obviously deviated from the data cluster, or from the equal value dash line. Data
points above the dash line indicate that CSES $h_mF_2$s are greater than the corresponding COSMIC
data, while data points below the dash line indicate a contrary situation that the COSMIC $h_mF_2$s are
greater than that of CSES. Larger errors are produced by these obviously deviated situations. In





spite of the data fluctuation, the nearly zero bias between the two sets, namely the mean data
differences, are so small that it can be neglected, which is in accord with the nearly normal
distribution of data differences as shown in Fig. 6b. The high correlation coefficient and the
normally distributed data differences again indicate that the overall $h_mF_2$ data of the two sets are
in a good agreement.
6       We also compare the daytime and nighttime $h_mF_2$s and the corresponding scatter plots are
given in Fig. 7. Correlation coefficient for daytime data is 0.9571, and for nighttime 0.8592. Similar
as $N_mF_2$, daytime $h_mF_2$ has a better correlation coefficient.
9       The mean data differences for daytime $h_mF_2$s is 0.62km with a RMSE of 10.17km; while the
mean data differences for nighttime $h_mF_2$s is 0.84 with a RMSE of 14.81km. The positive means of
data differences for both daytime and nighttime data indicate that the overall CSES $h_mF_2$s are
slightly greater than that of the COSMIC, but they are so small and can be neglected. The greater
RMSE of the nighttime data indicates an obvious more fluctuating nighttime $h_mF_2$s comparing to
the daytime $h_mF_2$s.

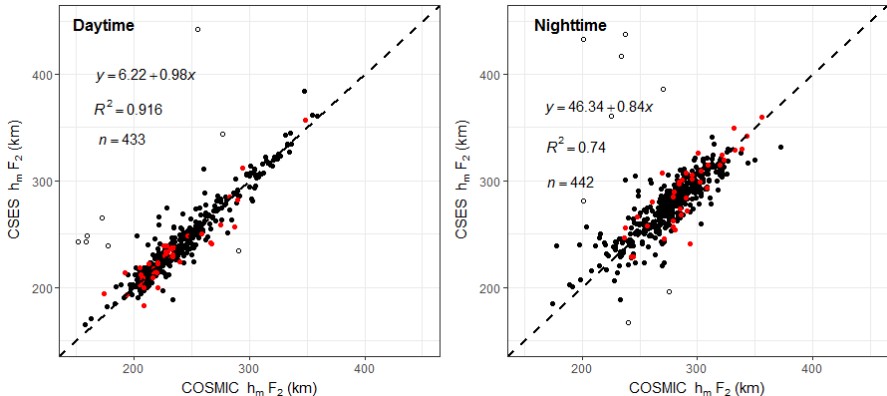

16                              Fig. 7 Scatter plot of $h_mF_2$ for daytime and nighttime data
(the dash line in Fig. 5 is the equal values line with a slope of 1)
The bias and RMSE for overall, daytime and nighttime data are given in Table 2 for a comparison.
19                              Table 2 Absolute error of $h_mF_2$ between CSES and COSMIC

|           | Correlation coefficient | Mean (km) | RMSE (km) |
|-----------|-------------------------|-----------|-----------|
| Total     | 0.9379                  | 0.73      | 13.02     |
| Daytime   | 0.9571                  | 0.62      | 10.17     |
| Nighttime | 0.8592                  | 0.84      | 14.81     |

From the results shown in Table 2 and Table 1, it can be seen that correlation of $N_mF_2$ is better
than that of $h_mF_2$ between the two sets. This result is in accord with the conclusion that the RO
measurements were better in $N_mF_2$ than in $h_mF_2$ (Chuo et al., 2011). Another point is that daytime
$h_mF_2$s are in better agreement than the nighttime data, which is similar as that of $N_mF_2$ data.
The overall comparison results of $h_mF_2$ are very good when comparing to prior validation
studies using ionsondes observations. Chuo et al. (2013) reported an $h_mF_2$ agreement about 0.87
using observations in low latitude south hemisphere from May 2006 to April 2008. Krankowski et
al. (2011) got a correlation coefficient of 0.949 when comparing COSMIC $h_mF_2$ data observed in
2008 with that from ionosondes in European mid-latitudes. Our result is consistent with their





results, which further proves that CSES $h_mF_2s$ are consistent and reliable with ionsondes
observations.
Krankowski et al. (2011) also obtained a bias of 2.8km and a standard deviation of 11.5km for
the COSMIC $h_mF_2$ data. Cherniak and Zakharenkova (2014) showed that COSMIC $h_mF_2s$ were in a
good agreement with Kharkov ISR observations of different seasons in 2008-2009, and bias and
standard deviations are less than 24 km and 29 km respectively. Habarulema et al. (2014) obtained
an error limit about 30km when comparing COSMIC $h_mF_2s$ with mid-latitude ionosonde using data
in 2008. Yue et al. (2011) suggested that the retrieval uncertainty in $h_mF_2$ is about 10km for COSMIC
simulation analysis. The nearly zero bias and the small RMSE between $h_mF_2$ of CSES and COSMIC
demonstrate that F region peak height parameter obtained by CSES and COSMIC are extremely
similar with each other, we therefore deduce that error between CSES $h_mF_2$ and ionsondes $h_mF_2$ is
comparable to prior results according to error propagation rules.
As a result, the significant correlation coefficient indicate the consistent variations between
CSES $h_mF_2s$ and $h_mF_2s$, and the nearly zero bias and the comparable error limits to prior studies
further indicate that CSES RO inverted $h_mF_2s$ are reliable considering the reliability of COSMIC RO
data validated by previous studies.

## 3.3 Comparison of EDPs

Besides the two most important parameters $N_mF_2$ and $h_mF_2$, electron density profiles (EDPs) are
also very important because EDPs can provide electron densities at different altitudes to depict
ionospheric 3D images from the bottom of ionosphere to the altitude of LEO satellite.
As EDPs from CSES and COSMIC have different altitudes due to the different satellite altitudes
of the two missions, only data under the altitude of the CSES satellite can be compared from the
co-located profiles. We therefore compare the inversed data at some special altitudes as the
numbers of data points are not identical for each matched profile pairs, and altitudes of each
inversed data are not identical for the two co-located profile pairs either.
For each altitude specified in section 2.3, we calculate the correlation coefficients using all the
matched data points at that altitude and the results are given in Table 3. Fig.8 gives the scatter
plots of all these altitudes, and data obtained in geomagnetic date are shown in red points, also
shown in the figure are the linear fitting equations, goodness-of-fit coefficients, and numbers of
data points involved in the calculation. Outliers are eliminated from the data sets using the same
criteria mentioned above.
Table 3 Correlation coefficients and RMSEs for the data at different altitudes of the profiles

| Altitude (km) | Correlation Coefficient | Absolute error | | Relative error | |
|---|---|---|---|---|---|
| | | Mean data difference | RMSE | Mean relative data differences | Relative RMSE |
| 500 | 0.9749 | $-0.01982\times10^5$ | $0.8824\times10^5$ | -1.716% | 35.90% |
| 450 | 0.9882 | $-0.01551\times10^5$ | $0.1070\times10^5$ | -0.6894% | 27.30% |
| 400 | 0.9929 | $-0.01923\times10^5$ | $0.1314\times10^5$ | -0.5888% | 20.29% |
| 350 | 0.9927 | $-0.02274\times10^5$ | $0.1946\times10^5$ | 0.7397% | 23.45% |
| 300 | 0.9908 | $-0.01881\times10^5$ | $0.2700\times10^5$ | 1.893% | 25.16% |
| 250 | 0.9874 | $-0.03198\times10^5$ | $0.3309\times10^5$ | 4.698% | 61.29% |
| 200 | 0.9691 | $-0.01090\times10^5$ | $0.3909\times10^5$ | 25.83% | 133.8% |



| 150 | 0.9564 | $-0.03161 \times 10^5$ | $0.2958 \times 10^5$ | 43.28% | 324.7% |
| 100 | 0.8883 | $-0.02330 \times 10^5$ | $0.2611 \times 10^5$ | 78.40% | 519.0% |

All the correlation coefficients in Table 3 can pass the significance test of confidence level 0.01,
which means that data points at different altitudes are highly correlated. When combining all the
results together, we can deduce that the co-located profiles from CSES and COSMIC sets are quite
similar to each other in spite of the global distribution of these profile pairs as shown in Fig. 2 in
Section 2.2. According to some studies, COSMIC profiles are in very good agreement with
observations from different ISRs (Lei et al., 2007; Kelley et al., 2009; Cherniak and Zakharenkova,
2014). Pedatella et al. (2015) compared COSMIC RO data at different altitudes with in situ
observations from CHAMP and C/NOSF and obtained the correlation coefficients are greater than
0.90, proving the consistency of the COSMIC profiles with in situ satellite observations. Based on
the high consistency between CSES and COSMIC profiles and prior COSMIC validation studies, we
can deduce that CSES profiles are generally agree with ISRs profiles according to similarity transitive
rules mentioned above (Langford et al., 2011).

13       Schreiner et al. (2007) showed that RMS is about $10^3/cm^3$ between 150 to 500km altitude,
whereas below 150km the RMS increases to a maximum of about $3 \times 10^3/cm^3$ at about 100km,
when comparing the RO profiles from different COSMIC satellites within 5km distance. Comparing
COSMIC profiles with ISR observations, Lei et al. (2007) suggested inversed errors are larger than
$10^5/cm^3$ at altitudes below ~150km and Cherniak and Zakharenkova (2014) obtained an error range
of $12\text{-}16 \times 10^4/cm^3$. Pedatella et al. (2015) obtained an overall bias of $0.22 \times 10^5/cm^3$ with a
standard deviation of $0.65 \times 10^5/cm^3$, and relative bias and standard deviation are 14.9% and 10.4%
respectively, when validating COSMIC data at different altitudes using CHAMP in situ observations;
they also compared COSMIC data with C/NOFS in situ observations, and got a relative bias of 5.6%
with a standard deviation 12.4%. They attributed the better agreement with in situ observations
from C/NOFS to the higher altitude of this satellite. Both the absolute and relative errors, as well
as error variation with altitudes shown in Table 3, are in accord with those studies, suggesting that
the CSES EDPs are reliable and within general error limits due to the high similarity and consistency
between CSES and COSMIC EDPs.

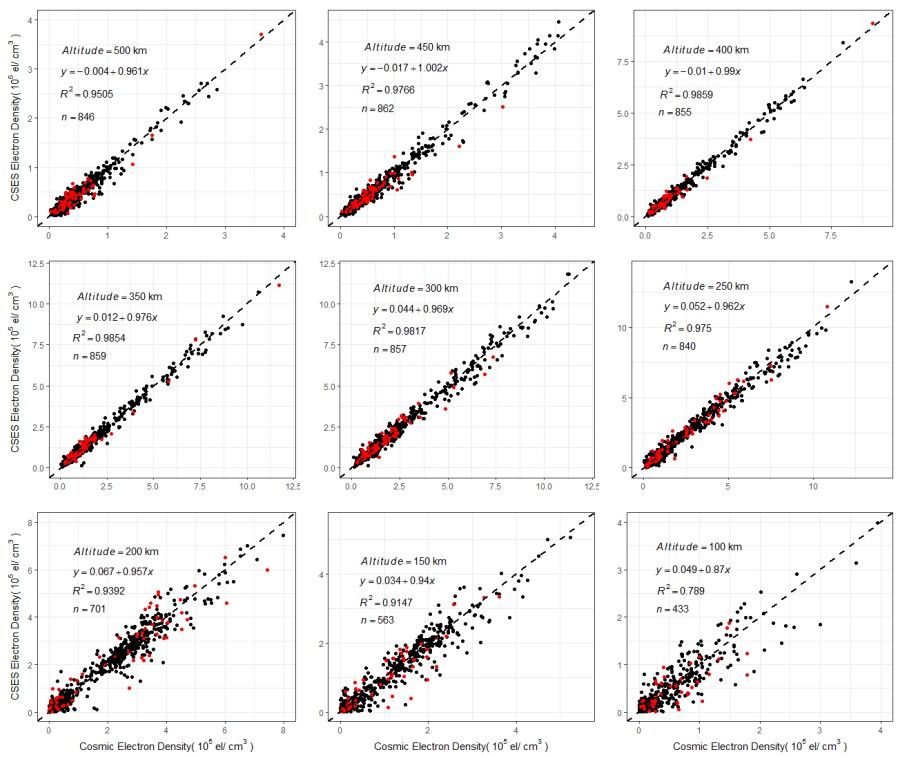

Fig. 8 Scatter plots of data from matched profiles at different altitudes
(the dash line in Fig. 5 is the equal values line with a slope of 1)
From the correlation coefficients given in Table 3, it can be seen that correlation coefficients
above 200 km are obviously greater than those below this altitude. The absolute mean differences
at different altitudes are comparable to each other. However, relative differences at different
altitudes are quite different; relative mean differences above 200km are extremely small, while
relative mean differences below this altitude (include this altitude) increase dramatically. We
obtained from Fig. 5 that the peak heights $h_mF_2$ of most profiles are located between 200km to
350km, the obviously high correlation coefficients in these regions indicate that RO inversed data
at and above peak height are more consistent with each other, whereas discrepancies between the
two data sets below the peak regions are much larger. This can be explained by the distribution
characteristics of the different ionospheric layers, and by the spherical assumption used in Abel
inversion method. As we know, electron density fluctuations in regions above the F2 peak become
smaller under geomagnetic quiet conditions if comparing with that at lower altitudes due to the
relative lower density according to electron density attenuation rules, it is therefore easier to
satisfy the spherical symmetry assumption when using the Abel inversion method in this region.
This spherical symmetry assumption is by far the most significant error source in the retrieval of
the electron density profiles (Lei et al., 2007). In addition, a shorter propagating distance in the
topside ionosphere for the radio signals from GPS to LEO will lead to a smaller error of straight line
propagation assumption. As suggested by Liu et al. (2010) that COSMIC RO can obtain reasonable
correct electron densities around and above F2 peak; however, assumption of spherical symmetry



introduces artificial plasma cave and plasma tunnel structures as well as electron density
enhancement at the geomagnetic equator at and below 250 km altitude, which will enlarge data
discrepancies as shown in Table 3. Syndergaard et al. (2006) also suggested larger errors at the
bottom of the retrieved profiles. The results shown in Table 3 in this study are in accord with those
studies, demonstrating that CSES EDPs have larger errors for data below 200km altitude, which is
similar as that of COSMIC.

7       An obvious characteristic shown in Table 3 is that all the means of data differences are negative
values though they are very small compare to the original measurements, which means the overall
CSES data at different altitudes are smaller than the corresponding COSMIC data. The all negative
mean data differences at different altitudes may indicate a possible systematic bias between the
two measurements. This systematic lower values at all altitudes is most likely caused by the first-
order estimation of the electron density at the altitude of the CSES satellite, rather than the spatial
differences of the co-located profile pairs, because spatial differences lead to random errors.
However, further confirmation of this error sources is required. It is also necessary to point out that
the signs of the mean relative data differences at altitudes ⩾400km are negative, similar as the
signs of the corresponding absolute errors; whereas the signs of the mean relative data differences
at altitudes below 400km are positive, just on the contrary to the signs of absolute mean data
differences. Further analysis shows that the opposite signs are caused by points where CSES data
are much larger than that of COSMIC, and thus lead to extremely larger relative errors, which
further indicates that data below the peak regions, especially below about 150km, fluctuate more
violently.
Besides spherical symmetry and straight line propagation assumptions, the larger discrepancies
at altitudes below peak regions can be explained by the different spatial locations of the matched
profiles. Although the peak values of co-located profile pairs are near each other according to
selection criteria, data points other than peak values on the matched profile pairs may exceed the
selection criteria and result in larger distances due to the different tangent point path of the
matched profile pairs. As a result, a larger distance will lead to larger discrepancy between the
corresponding data sets. In addition, the tangent point path of the matched profiles may have
different directions, which will lead to different inversion results because each inversed data
represents average electron densities along the radio ray path. In regions with large horizontal
gradients, the different ray path can cause obvious difference between the matched profiles. At
altitudes below 200km, especially below 150 km, sporadic E-layers can cause large horizontal
gradients, and then lead to large inversion error. Wu et al. (2009) suggested that the large relative
error below 150 km is due to the errors transferred from upper altitude (the F layer) and the very
small electron density at that altitude. They also suggested that the larger ray separations can
induce larger errors which can be transferred to low altitudes; phase measurement errors induce
small relative fluctuations on the electron density at the topside ionosphere, but can cause large
relative fluctuations at low altitude ionosphere, because small electron density at low altitude is
sensitive to the phase errors. It is therefore concluded that many sources can cause large errors for
measurements at altitudes below 150km, which as a result lead to the large discrepancies between
CSES and COSMIC RO data at the bottom of the ionosphere.
Based on the above analysis, we conclude that CSES profiles are generally consistent with
COSMIC data very well and are reliable for data applications. However, larger discrepancies are
found at lower altitudes between the two sets comparing to data differences at higher altitudes.



Therefore, special attention should be paid to data below 200km in applications due to the relative
large discrepancies between the two datasets.
**4.    Summary and Conclusions**
Validation of the CSES RO data is carried out to estimate the consistency and reliability of the
CSES RO data using the globally distributed measurements from the COSMIC mission covering the
date range from February 12, 2018 to March 31, 2019 as COSMIC RO data have been widely
validated their consistency and reliability using data from different measurements in global scale.
Comparing CSES $N_mF_2$, $h_mF_2$, and EDP data at some special altitudes, with corresponding COSMIC
RO data, we obtain the following results.
(1) CSES $N_mF_2$ data are highly consistent with that from COSMIC with a correlation coefficient

11        of 0.9891. The mean data differences is $0.01235 \times 10^5/cm^3$ with a RMSE of $0.3680 \times$

12        $10^5/cm^3$; the relative mean differences is 2.1% with a relative RMSE of 16.4%. Correlation

13        between daytime $N_mF_2$ data is obviously better than that of nighttime $N_mF_2$ data.

(2) CSES $h_mF_2$ data are also very consistent with COSMIC data, with a correlation coefficients

15        of 0.9379. The bias between the two sets is 0.73 km with a RMSE of 13.02km. Again,

16        daytime $h_mF_2$ has a better correlation than nighttime data.

(3) Co-located profiles between CSES and COSMIC are generally consistent with each other

18        very well, with a better agreement for data at and above peak height regions (200km)

19        than those below this regions. For EDP data below 200 km altitude, special attention

20        should be paid due to the relative large discrepancies between the two sets.

(4) Based on the results of comparing COSMIC data with data from different measurements,

22        it is deduced that CSES RO data are within the error limits obtained by previous studies

23        according to error propagation rules.

GOX payload onboard CSES satellite can obtain over 500 occultation events each day, which
provide a large dataset for the study of 3D distribution of the ionospheric electron density when
combining with the in situ electron density measurements obtained by LAP onboard CSES. The
relatively thorough comparison work in this paper demonstrates that the CSES RO data are
consistent very well with the corresponding COSMIC data, proving that the CSES RO data are
reliable for applications on ionospheric-related problems. However, as many RO related studies
suggest that asymmetry of electron density distribution is the main source of the Abel inversion
transformation (Schreiner et al., 1999; Syndergaard et al., 2006; Lei et al., 2007), and this inversion
error varies with solar activity, season, geomagnetic latitude and local time (Wu et al., 2009). The
CSES RO data in this study cover all the latitudes and four seasons with fixed local time under lower
solar activity condition, and solar activity in this study is similar as most of the COSMIC validation
studies, the comparison results will therefore applicable to data with similar low solar activity
conditions. More subsequent validation work will be conducted and presented using data
accumulated under different solar activities.



**Acknoledgement**
COSMIC Radio Occultation data were downloaded from ftp://cdaac-
ftp.cosmic.ucar.edu/cosmic/level2 /ionPrf/. This research was supported by the National Key R&D
Program of China (Grant no. 2018YFC1503505), and by the Institute of Crustal Dynamics, China
Earthquake Administration (Grant no. ZDJ2018-18).

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
