# Peer review of "Comparison of CSES ionospheric RO data with COSMIC measurements"

_Annales Geophysicae, 2019_

## Referee Comment (RC1) · Anonymous Referee #1 · 25 Jul 2019

The paper is devoted to a comparison of Radio Occultation (RO) observations from newly launched CSES satellite to analogist COSMIC RO observations. The observations demonstrate an excellent coincidence for NmF2 and hmF2 and this may be considered as an important result. The main scientific objective is declared to monitor earthquake related disturbances in the ionosphere. Keeping in mind good results of the undertaken comparison the authors have concluded that the CSES RO data are applicable for most ionospheric-related studies. There are no objections concerning the comparison itself and a good coincidence with COSMIC observations indicate that CSES provides reliable observations at the level of COSMIC. This may be considered as the main result of the paper. However whether RO observations provide real Ne(h) profiles which may be used for ionospheric studies and applications is a question. The

authors give a long list of papers confirming the reality of RO Ne(h) profiles as well as NmF2 and hmF2 read from such profiles. However there are publications expressing doubts in relation with RO Ne(h) observations and comparisons with ground-based ionosonde and ISR data tell us the situation with RO is not that straightforward. The main problem is the spherical symmetry assumption used in the method which is not valid in many real situations: the periods of sunrise and sunset, low latitude and sub-auroral ionosphere where equatorial crests and the main ionospheric trough create spatial asymmetry. They have absolutely correctly selected 1400LT and 0200LT for their comparisons as these periods are the most stable in the ionosphere and the spatial asymmetry is supposed to be the least. However any monitoring implies 24-hour observations and problems will appear inevitably. A more extended comparison of COSMIC RO profiles with Arecibo ISR observations in June 2006 was presented by Kelly et al. (2009). This comparison included 32 profiles in overall, which were obtained by using the Abel transform method that was developed in two versions independently at UCAR and JPL. An interesting result is that the two methods of Abel transformation may give quite different EDP telling us that the data processing methods are not straightforward. Both NmF2 and hmF2 scatterplots (their Fig. 10 and 11) manifest large scatter, but NmF2 is seen to be determined better than hmF2. The SD of the former is 1x105 cm-3, while no statistical results are given for hmF2. It is pointed out that the best agreement between hmF2 RO values with the ISR ones was obtained near 300 km, but RO values are lower than ISR values below this height and higher than ISR values above it.

Similar but more representative results can be found in Tsai et al. (2009) who made a comparison of COSMIC observations with 49 worldwide distributed ionosondes for the period of 20 June-27 September, 2006. Their Fig. 6 exhibits a large scatter of COSMIC foF2 versus ionosonde values with mean relative deviation 20% (40 % in NmF2). This coincides with earlier estimates by Schreiner et al. (1999) and Tsai et al. (2001). A comparison on hmF2 has shown that "the FS3/COSMIC hmF2s do not coincide well with the ionosonde hmF2s". Their Fig. 9 shows that COSMIC hmF2s are much lower

than ionosonde values.

Schreiner et al. (2007) analyzing the precision of GPS RO from FORMOSAT-3/COSMIC mission concluded: "Thus retrieved electron density profiles are expected to have rather poor accuracy when interpreted as actual vertical profiles". According to Schreiner et al. (2007) the largest error in the GPS IRO retrieved electron density profiles is due to strong horizontal gradients disrupting the assumption of spherical symmetry. This assumption, in cases with large NmF2 values, can result in either positive or negative errors larger than 105 cm-3 at the bottom of the retrieved profiles [Syndergaard et al., 2006].

Summarizing the results of the previous comparisons between RO Ne(h) observations with both ionosonde and ISR measurements it is possible to conclude the following. The agreement between RO retrieved and ionosonde foF2 values is within 10-20% (20-40% in NmF2).

Coming back to the reviewed paper and considering it in the light of its title "A Comparison..." I may conclude that this is an interesting paper where good results were obtained. It is clearly written and well-organized. After some minor corrections it may be recommended for publication in AG.

Specific comments

L8 to inverse electron density related parameters. Maybe to retrieve of to infer

L13,17 what is NmF2s and hmF2s ? Not explained

L30 for both the 3-D earthquake observation and geophysical field measurement

Poor phrase. On one hand the authors can only speak about possible ionospheric precursors of earthquakes as this is still only a suggestion statistically not confirmed, moreover having two observations per day at a given location hardly one can follow the development in time of the earthquake preparation process. On the other hand what does mean "geophysical field measurement"? Which parameters are meant?

[Figure]

P2 L32 show that the CSES RO NmF2 data are generally consistent with data from other measurements. What does this mean "consistent" to which extent? – a quantitative estimate should be given.

P2 L38 from vertical ionosondes Incorrect term. "Ionospheric vertical sounding" exists.

P2 L41 most located on inland and fewer in the oceans Ground-based ionosondes of ionospheric vertical sounding are located on the continents and islands in the oceans but not in the oceans.

P3 L15 to check the consistency and reliability of CSES profiles except for NmF2 and hmF2 parameters.

How to understand this phrase? NmF2 and hmF2 are the main ionospheric parameters and the authors do not want to check them?

P3 L27 to inverse Maybe to retrieve or to infer

P5 L6 it should be pointed out here that RO retrieved electron density profiles cannot be interpreted as actual vertical profiles because both the LEO and GPS are in motion during the occultation process. If RO Ne(h) profiles are not real ionospheric electron density profiles (this has been formulated by Schreiner et al. (2007) then how these RO observations can be used for ionospheric investigations? If RO Ne(h) profiles are not real then any comparisons with ground-based NmF2 and hmF2 observations are senseless.

P6 L6 The ionospheric spatial correlation distance depends on geophysical conditions (McNamara, 2009) - solar minimum or maximum, magnetically quiet or disturbed conditions. At middle latitudes for practical applications during quiet conditions may be used ïA¿ 500 km in latitudes and ïA¿ (700-1000) km in longitude. So 2 x 6 deg correlation distances selected by the authors may be considered as reasonable.

P5 29 NmF2 appearing below 150km

The height of 150 km is unreal for F2-layer. Even under the deep minimum of solar activity in 2008-2009 daytime hmF2 according to ISR observations never was lower than 200-210 km. Therefore all RO hmF2 lower than 200 km should be considered as erroneous ones.

P6 L7 the over a year temporal segment Poor style

P6 L12 Special attentions should be paid on the local time issue when CSES and COSMIC RO data are combined together. The phrase is not clear.

P6 L29 The RO Ne(h) profiles very often are not smooth at all. How such cases were developed?

P7 P13 also L13 as hollow circles Open circles

P8 L15 This is usual MRD (mean relative deviation) –there is no need to invent new definitions

P9 Table 1 Hardly real accuracy of NmF2 determination requires 5 digits.

P9 L27 There is another point to point out. Poor style.

P10 L6 CSES RO derived peak values are in very good agreement with COSMIC and ground based measurements. No comparisons with ground-based NmF2 observations are done in the paper.

P10 L 28 As we know, the nighttime data has a more complex spatial distribution pattern compare to daytime data although daytime data are affected by solar radiation during day time.

Not "although" but namely due to solar ionization NmF2 variations are smoother during daytime hours.

P10 L36 They suggested that COSMIC measurements are acceptable under geomagnetic disturbed conditions when comparing COSMIC RO data with observations from

Sanya, a lower latitude ionosonde in China. Hardly one can agree with this statement. The equatorial anomaly introduces a spatial asymmetry especially during storm periods. This asymmetry should affect RO results.

Part 3.2 and Table 2 indicate excellent results RMSE <15 km This is a difference within the RO method obtained by two similar devices. But it should be stressed that the difference between RO hmF2 and real hmF2 may be different. The most accurate hmF2 provide ISR observations and only such comparisons may give a real estimate of RO hmF2 accuracy.

---

## Author Comment (AC1) · 31 Jul 2019

First of all, we thank the reviewer for his careful reviewing of our paper, and for his suggestions on the improvement of this paper.

We think the reviewer completely get the idea behind this paper. As the reviewer points out, good correlation between CSES and COSMIC indicates "CSES provides reliable observations at the level of COSMIC". The reliability of COSMIC can be proved by many prior studies of comparing COSMIC observations with different measurements. Based on this logical relation and error propagation rules, we can deduced the reliability of the CSES observations.

As to the problems pointed out by the reviewer, we explain one by one.

(1)L8 to inverse electron density related parameters. Maybe to retrieve of to infer.

We will follow the suggestion, and modify the inappropriate word used in the abstract.

(2)L13,17 what is NmF2s and hmF2s ? Not explained

NmF2s and hmF2s are the plural forms. As we know, we obtained over 700 co-located RO events, we therefore can get over 700 NmF2 and hmF2 data points, and therefore plural forms are used here.

(3)L30 for both the 3-D earthquake observation and geophysical field measurement.

This expression is from the brochure of CSES. My understanding: There are different earthquake observation network systems distributed on the ground in China. CSES satellite is the first system to observe possible earthquake-related quantities from space. Combing the ground-based and space-based systems together, a 3-D observation system is formed.

Geophysical field measurement, as mentioned in the paper, there are 8 payloads on-board CSES. All of these observations can be regarded as the extent of geophysical observations on the ground. CSES has a short revisiting period, this ensures that the observations time intervals are short. Observations from many times circular orbits are helpful to create the background (field).

(4)P2 L32 show that the CSES RO NmF2 data are generally consistent with data from other measurements. What does this mean "consistent" to which extent? – a quantitative

This conclusion is from the paper referenced in our paper. There is no clearly quantitative conclusion and application suggestion in that paper. That is why we conduct the comparison work of this paper, and application suggestion is given based on our quantitative work. We think the quantitative comparison and application suggestion is very important before the CSES data is shared for scientific community.

(5)P2 L38 from vertical ionosondes Incorrect term. "Ionospheric vertical sounding" exists.

We will follow the suggestion, and modify the incorrect usage. Many thanks for the careful review work.

(6)P2 L41 most located on inland and fewer in the oceans. Ground-based ionosondes of ionospheric vertical sounding are located on the continents and islands in the oceans but not in the oceans.

We will modify this mistake in our paper.

(7)P3 L15 to check the consistency and reliability of CSES profiles except for NmF2 and hmF2 parameters. How to understand this phrase? NmF2 and hmF2 are the main ionospheric parameters and the authors do not want to check them?

'Except for' here means 'besides', both COSMIC and CSES have similar retrieved data, this convenience enables the validation of the NmF2 and hmF2. Moreover, it also enables the validation of RO profiles as well.

Sorry for the poor expression. We will improve our expression in the modification version.

(8)P3 L27 to inverse Maybe to retrieve or to infer

We accept the suggestion and will modify it.

(9)P5 L6 it should be pointed out here that RO retrieved electron density profiles cannot be interpreted as actual vertical profiles because both the LEO and GPS are in motion during the occultation process. If RO Ne(h) profiles are not real ionospheric electron density profiles (this has been formulated by Schreiner et al. (2007) then how these RO observations can be used for ionospheric investigations? If RO Ne(h) profiles are not real then any comparisons with ground-based NmF2 and hmF2 observations are senseless.

This sentence means that the RO profiles are not identical as the profiles obtained by ionosonde vertical sounding or by incoherent scatter radar. For the latter case, the observation point is fixed, the profile indicates the data observed at different altitude of the same point. For RO profile, the location of the profile is not fixed, therefore data from the same profile means different locations. We also draw the projection track of the profiles from the co-located profiles, as following Fig. 1.

Sorry for the inappropriate expression. We will try to improve the expression and make it clearer.

(10)P6 L6 The ionospheric spatial correlation distance depends on geophysical conditions (McNamara, 2009) - solar minimum or maximum, magnetically quiet or disturbed conditions. At middle latitudes for practical applications during quiet conditions may be used ïAËŻÂ£ 500 km in latitudes and ïAËŻÂ£ (700-1000) km in longitude. So 2 x 6 deg correlation distances selected by the authors may be considered as reasonable.

We also plot some figures to ascertain the selection of this correlation distances, as shown above is one of these figures. Many thanks to the reviewer to ascertain this work.

(11)P5 29 NmF2 appearing below 150km The height of 150 km is unreal for F2-layer. Even under the deep minimum of solar activity in 2008-2009 daytime hmF2 according to ISR observations never was lower than 200-210 km. Therefore all RO hmF2 lower than 200 km should be considered as erroneous ones.

We will eliminate the points where NmF2 appears below 200km according to the suggestion.

(12)P6 L7 the over a year temporal segment Poor style

Sorry for the poor English. We will improve the English of this paper during the modification work.

(13)P6 L12 Special attentions should be paid on the local time issue when CSES and

COSMIC RO data are combined together. The phrase is not clear.

As mentioned in the paper, the CSES local time is about 14:00 during the day or 2:00 during the night. RO data local time is around the two local time. Therefore, if we want to combine the data from the two missions, COSMIC data with similar CSES local time must be selected out. We will make this sentence clearer after the modification revision.

(14)P6 L29 The RO Ne(h) profiles very often are not smooth at all. How such cases were developed?

As the altitude of CSES is 507km, CSES RO data is below this altitude. These special altitudes are selected just for simple. To get the electron density data at these altitudes, we calculate the average density between Altitude-10km to Altitude+10km, as mentioned in the passage following the one mentioned by the reviewer. After this calculation, data fluctuation is erased and one single data is obtained. By this way we can compare all the data pairs at one altitude together, as it is shown in Section 3.3.

(15)P7 P13 also L13 as hollow circles Open circles

We accept the suggestion and will change it.

(16)P8 L15 This is usual MRD (mean relative deviation) –there is no need to invent new definitions.

We accept the suggestion. We will delete the equation and use MRD. Many thanks to the reviewer for this suggestion.

(17)P9 Table 1 Hardly real accuracy of NmF2 determination requires 5 digits.

This is not an indication of precision, but to maintain the same number of significant digits.

(18)P9 L27 There is another point to point out. Poor style.

Sorry for the poor English. We will improve the English of this paper during the modification work.

(19)P10 L6 CSES RO derived peak values are in very good agreement with COSMIC and ground based measurements. No comparisons with ground-based NmF2 observations are done in the paper.

This conclusion is based on the high consistency between COSMIC and CSES data, and on the COSMIC validation work conducted using ground-based ISR and ionosonde measurements, as discussed earlier in this passage. According to the error propagation rule and correlation transitive rule, we can get this conclusion.

(20)P10 L 28 As we know, the nighttime data has a more complex spatial distribution pattern compare to daytime data although daytime data are affected by solar radiation during day time. Not "although" but namely due to solar ionization NmF2 variations are smoother during daytime hours.

Sorry for the mistake usage of the conjunction. We will improve the English carefully.

(21)P10 L36 They suggested that COSMIC measurements are acceptable under geomagnetic disturbed conditions when comparing COSMIC RO data with observations from Sanya, a lower latitude ionosonde in China. Hardly one can agree with this statement. The equatorial anomaly introduces a spatial asymmetry especially during storm periods. This asymmetry should affect RO results.

Part 3.2 and Table 2 indicate excellent results RMSE <15 km This is a difference within the RO method obtained by two similar devices. But it should be stressed that the difference between RO hmF2 and real hmF2 may be different. The most accurate hmF2 provide ISR observations and only such comparisons may give a real estimate of RO hmF2 accuracy.

COSMIC measurements are acceptable under geomagnetic disturbed conditions. This conclusion is from the reference paper by Hu et al. (2014) using data from 2008-2013

in Sanya Station, China. This conclusion may need more validation work. However, since it is published, we can reference its conclusion.

We are completely agree with the reviewer's opinion that the most accurate hmF2 is from ISR. We are now collecting ISR data, validation of CSES RO data using ISR observations is under preparation.

————————————————————

2019-02-17, Night

Fig. 1.

---

## Referee Comment (RC2) · Anonymous Referee #2 · 11 Aug 2019

The scientific quality of the manuscript is good. The results are clear and well discussed. Some minor revisions are needed. The use of measurements of ionospheric parameters as seismic precursors is still a debated issue. In any case the focal point of the article is the comparison between CSES and COSMIC RO data and from this point of view the article is convincing. The analysis of the data was carried out with a rigorous method in accordance with the ANGEO standard. I suggest publication after minor revisions.

Line 30 The presismic character of observations from space is still under discussion. Please avoid over-stretched statements about precursors that are still supported by poor statistical confirmation.

Page 6 Line 7 Please review the style.

[Figure]

Page 6 Line 12 Please clarify the definition of the time at which CSES and COSMIC data are compared.

Page 9 Line 27 Change style.

In general, a revision of English is required.
* * *

---

## Author Comment (AC2) · 12 Aug 2019

First of all, we thank the referee #2 for his/her advices and suggestions on our work. Response to the suggestions are as following.

(1) Line 30 The presismic character of observations from space is still under discussion. Please avoid over-stretched statements about precursors that are still supported by poor statistical confirmation.

Response: As referee#2 points out, earthquake precursor (preseismic character) is still a debate. "Are there precursors helpful to earthquake prediction" is still a scientific problem. This paper is to validate the data obtained by the newly launched satellite. A general introduction to the satellite is necessary to give the readers complete information about the satellite. Therefore, a short introduction is given in this paragraph. The introduction is shortened from the CSES brochure. It is just an objective introduction, no mean to discuss problems with earthquake precursors.

(2) Page 6 Line 7 Please review the style.

We will improve the English of this paper during our subsequent modification work.

(3) Page 6 Line 12 Please clarify the definition of the time at which CSES and COSMIC data are compared.

The time criterion to select matching CSES and COSMIC event is defined in Page 5 Line 18-19: the time difference between the matching occultation pairs is less than 30 min.

Page 6 Line 12 stresses that COSMIC RO data covers all local time, while CSES RO data only covers 2 special local time due to its special orbit. Data users should pay attention to this.

(4) Page 9 Line 27 Change style.

We will improve the English of this paper during our subsequent modification work.

---

## Author Response (AR1)

**Response to referees**

**Comments from referees**

**Referee 1**

   The paper is devoted to a comparison of Radio Occultation (RO) observations from newly launched CSES satellite to analogist COSMIC RO observations. The observations demonstrate an excellent coincidence for NmF2 and hmF2 and this may be considered as an important result. The main scientific objective is declared to monitor earthquake related disturbances in the ionosphere. Keeping in mind good results of the undertaken comparison the authors have concluded that the CSES RO data are applicable for most ionospheric-related studies. There are no objections concerning the comparison itself and a good coincidence with COSMIC observations indicate that CSES provides reliable observations at the level of COSMIC. This may be considered as the main result of the paper. However whether RO observations provide real Ne(h) profiles which may be used for ionospheric studies and applications is a question. The authors give a long list of papers confirming the reality of RO Ne(h) profiles as well as NmF2 and hmF2 read from such profiles. However there are publications expressing doubts in relation with RO Ne(h) observations and comparisons with ground-based ionosonde and ISR data tell us the situation with RO is not that straightforward. The main problem is the spherical symmetry assumption used in the method which is not valid in many real situations: the periods of sunrise and sunset, low latitude and subauroral ionosphere where equatorial crests and the main ionospheric trough create spatial asymmetry. They have absolutely correctly selected 1400LT and 0200LT for their comparisons as these periods are the most stable in the ionosphere and the spatial asymmetry is supposed to be the least. However any monitoring implies 24-hour observations and problems will appear inevitably. A more extended comparison of COSMIC RO profiles with Arecibo ISR observations in June 2006 was presented by Kelly et al. (2009). This comparison included 32 profiles in overall, which were obtained by using the Abel transform method that was developed in two versions independently at UCAR and JPL. An interesting result is that the two methods of Abel transformation may give quite different EDP telling us that the data processing methods are not straightforward. Both NmF2 and hmF2 scatterplots (their Fig. 10 and 11) manifest large scatter, but NmF2 is seen to be determined better than hmF2. The SD of the former is 1x105 cm-3, while no statistical results are given for hmF2. It is pointed out that the best agreement between hmF2 RO values with the ISR ones was obtained near 300 km, but RO values are lower than ISR values below this height and higher than ISR values above it.

   Similar but more representative results can be found in Tsai et al. (2009) who made a comparison of COSMIC observations with 49 worldwide distributed ionosondes for the period of 20 June-27 September, 2006. Their Fig. 6 exhibits a large scatter of COSMIC foF2 versus ionosonde values with mean relative deviation 20% (40 % in NmF2). This coincides with earlier estimates by Schreiner et al. (1999) and Tsai et al. (2001). A comparison on hmF2 has shown that "the FS3/COSMIC hmF2s do not coincide well with the ionosonde hmF2s". Their Fig. 9 shows that COSMIC hmF2s are much lower than ionosonde values.

   Schreiner et al. (2007) analyzing the precision of GPS RO from FORMOSAT-3/COSMIC mission concluded: "Thus retrieved electron density profiles are expected to have rather poor

accuracy when interpreted as actual vertical profiles". According to Schreiner et al. (2007) the largest error in the GPS IRO retrieved electron density profiles is due to strong horizontal gradients disrupting the assumption of spherical symmetry. This assumption, in cases with large NmF2 values, can result in either positive or negative errors larger than 105 cm-3 at the bottom of the retrieved profiles [Syndergaard et al., 2006].

Summarizing the results of the previous comparisons between RO Ne(h) observations with both ionosonde and ISR measurements it is possible to conclude the following. The agreement between RO retrieved and ionosonde foF2 values is within 10-20% (20-40% in NmF2).

Coming back to the reviewed paper and considering it in the light of its title "A Comparison: : :" I may conclude that this is an interesting paper where good results were obtained. It is clearly written and well-organized. After some minor corrections it may be recommended for publication in AG.

Specific comments

L8 to inverse electron density related parameters. Maybe to retrieve of to infer

L13,17 what is NmF2s and hmF2s ? Not explained

L30 for both the 3-D earthquake observation and geophysical field measurement Poor phrase. On one hand the authors can only speak about possible ionospheric precursors of earthquakes as this is still only a suggestion statistically not confirmed, moreover having two observations per day at a given location hardly one can follow the development in time of the earthquake preparation process. On the other hand what does mean "geophysical field measurement"? Which parameters are meant?

P2 L32 show that the CSES RO NmF2 data are generally consistent with data from other measurements. What does this mean "consistent" to which extent? – a quantitative estimate should be given.

P2 L38 from vertical ionosondes Incorrect term. "Ionospheric vertical sounding" exists.

P2 L41 most located on inland and fewer in the oceans Ground-based ionosondes of ionospheric vertical sounding are located on the continents and islands in the oceans but not in the oceans.

P3 L15 to check the consistency and reliability of CSES profiles except for NmF2 and hmF2 parameters.

How to understand this phrase? NmF2 and hmF2 are the main ionospheric parameters and the authors do not want to check them?

P3 L27 to inverse Maybe to retrieve or to infer

P5 L6 it should be pointed out here that RO retrieved electron density profiles cannot be interpreted as actual vertical profiles because both the LEO and GPS are in motion during the occultation process. If RO Ne(h) profiles are not real ionospheric electron density profiles (this has been formulated by Schreiner et al. (2007) then how these RO observations can be used for ionospheric investigations? If RO Ne(h) profiles are not real then any comparisons with ground-based NmF2 and hmF2 observations are senseless.

P6 L6 The ionospheric spatial correlation distance depends on geophysical conditions (McNamara, 2009) - solar minimum or maximum, magnetically quiet or disturbed conditions. At middle latitudes for practical applications during quiet conditions may be used ïA¿ 500 km in latitudes and ïA¿ (700-1000) km in longitude. So 2 x 6 deg correlation distances selected by the authors may be considered as reasonable.

P5 29 NmF2 appearing below 150km. The height of 150 km is unreal for F2-layer. Even under the deep minimum of solar activity in 2008-2009 daytime hmF2 according to ISR observations never was lower than 200-210 km. Therefore all RO hmF2 lower than 200 km should be considered as erroneous ones.

P6 L7 the over a year temporal segment Poor style

P6 L12 Special attentions should be paid on the local time issue when CSES and COSMIC RO data are combined together. The phrase is not clear.

P6 L29 The RO Ne(h) profiles very often are not smooth at all. How such cases were developed?

P7 P13 also L13 as hollow circles Open circles

P8 L15 This is usual MRD (mean relative deviation) –there is no need to invent new definitions

P9 Table 1 Hardly real accuracy of NmF2 determination requires 5 digits.

P9 L27 There is another point to point out. Poor style.

P10 L6 CSES RO derived peak values are in very good agreement with COSMIC and ground based measurements. No comparisons with ground-based NmF2 observations are done in the paper.

P10 L 28 As we know, the nighttime data has a more complex spatial distribution pattern compare to daytime data although daytime data are affected by solar radiation during day time.
Not "although" but namely due to solar ionization NmF2 variations are smoother during daytime hours.

P10 L36 They suggested that COSMIC measurements are acceptable under geomagnetic disturbed conditions when comparing COSMIC RO data with observations from Sanya, a lower latitude ionosonde in China. Hardly one can agree with this statement. The equatorial anomaly introduces a spatial asymmetry especially during storm periods. This asymmetry should affect RO results. Part 3.2 and Table 2 indicate excellent results RMSE <15 km This is a difference within the RO method obtained by two similar devices. But it should be stressed that the difference between RO hmF2 and real hmF2 may be different. The most accurate hmF2 provide ISR observations and only such comparisons may give a real estimate of RO hmF2 accuracy.

**Referee 2**

The scientific quality of the manuscript is good. The results are clear and well discussed. Some minor revisions are needed. The use of measurements of ionospheric parameters as seismic precursors is still a debated issue. In any case the focal point of the article is the comparison between CSES and COSMIC RO data and from this point of view the article is convincing. The analysis of the data was carried out with a rigorous method in accordance with the ANGEO standard. I suggest publication after minor revisions.

Line 30 The presismic character of observations from space is still under discussion. Please avoid over-stretched statements about precursors that are still supported by poor statistical confirmation.

Page 6 Line 7 Please review the style.

Page 6 Line 12 Please clarify the definition of the time at which CSES and COSMIC data are compared.

Page 9 Line 27 Change style.

In general, a revision of English is required.

**Authors' response**

**Response to Referee #1**

First of all, we thank the reviewer for his careful reviewing of our paper, and for his suggestions on the improvement of this paper.

We think the reviewer completely get the idea behind this paper. As the reviewer points out, good correlation between CSES and COSMIC indicates "CSES provides reliable observations at the level of COSMIC". The reliability of COSMIC can be proved by many prior studies of comparing COSMIC observations with different measurements. Based on this logical relation and error propagation rules, we can deduced the reliability of the CSES observations.

As to the problems pointed out by the reviewer, we explain one by one.

(1) L8 to inverse electron density related parameters. Maybe to retrieve of to infer.

We accept the suggestion, and use retrieve instead of inverse all over the paper.

(2) L13,17 what is NmF2s and hmF2s ? Not explained

Sorry for this unclearness. NmF2s and hmF2s are the plural forms. As we know, we obtained over 700 co-located RO events, we therefore can get over 700 NmF2 and hmF2 data points, and therefore plural forms are used here.

Since this usage may cause confusion, we change to the original form in the abstract section, but remain plural forms in the text.

(3) L30 for both the 3-D earthquake observation and geophysical field measurement

This expression is from the brochure of CSES. My understanding: There are different earthquake observation network systems distributed on the ground in China. CSES satellite is the first system to observe possible earthquake-related quantities from space. Combing the ground-based and space-based systems together, a 3-D observation system is formed.

Geophysical field measurement, as mentioned in the paper, there are 8 payloads onboard CSES. All of these observations can be regarded as the extent of geophysical observations on the ground. CSES has a short revisiting period, this ensures that the observations time intervals are short. Observations from many times circular orbits are helpful to create the background (field).

Since the background of this sentence is a long story, the paper won't explain it in detail. Only a general description is added to give the readers some information about this satellite. Therefore, only a little improvement is made here.

(4) P2 L32 show that the CSES RO NmF2 data are generally consistent with data from other measurements. What does this mean "consistent" to which extent? – a quantitative

This conclusion is from the paper, Cheng et al. (2018), referenced in our paper. There is no clearly quantitative conclusion and application suggestion in that paper. That is why we conduct the comparison work of this paper, and application suggestion is given based on our quantitative work. We think the quantitative comparison and application suggestion is very important before the CSES data is shared for scientific community.

To make it clear, we make some improvements here.

(5) P2 L38 from vertical ionosondes Incorrect term. "Ionospheric vertical sounding" exists.

We accept the suggestion, and modify the incorrect usage.

(6) P2 L41 most located on inland and fewer in the oceans. Ground-based ionosondes of ionospheric vertical sounding are located on the continents and islands in the oceans but not in the oceans.

We made a little improvement of this sentence to make it clear.

(7) P3 L15 to check the consistency and reliability of CSES profiles except for NmF2 and hmF2 parameters.

How to understand this phrase? NmF2 and hmF2 are the main ionospheric parameters and the authors do not want to check them?

'Except for' here means 'besides', both COSMIC and CSES have similar retrieved data, this convenience enables the validation of the NmF2 and hmF2. Moreover, it also enables the validation of RO profiles as well.

We modify the sentence to make it easy to understand.

(8) P3 L27 to inverse Maybe to retrieve or to infer

We accept the suggestion, and modify the sentence.

(9) P5 L6 it should be pointed out here that RO retrieved electron density profiles cannot be interpreted as actual vertical profiles because both the LEO and GPS are in motion during the occultation process. If RO Ne(h) profiles are not real ionospheric electron density profiles (this has been formulated by Schreiner et al. (2007) then how these RO observations can be used for ionospheric investigations? If RO Ne(h) profiles are not real then any comparisons with ground-based NmF2 and hmF2 observations are senseless.

This sentence means that the RO profiles are not identical as the profiles obtained by ionosonde vertical sounding or by incoherent scatter radar. For the latter case, the observation point is fixed, the profile indicates the data observed at different altitude of the same point. For RO profile, the location of the profile is not fixed, therefore data from the same profile means different locations. We also draw the projection track of the profiles from the co-located profiles, as following.

To make it clear, we make some improvement here, and a detailed description is given.

(10) P6 L6 The ionospheric spatial correlation distance depends on geophysical conditions (McNamara, 2009) - solar minimum or maximum, magnetically quiet or disturbed conditions. At middle latitudes for practical applications during quiet conditions may be used ïA¿ 500 km in latitudes and ïA¿ (700-1000) km in longitude. So 2 x 6 deg correlation distances selected by the authors may be considered as reasonable.

We also plot some figures to ascertain the selection of this correlation distances. Many thanks to the reviewer to affirmation of this work.

(11) P5 29 NmF2 appearing below 150km

The height of 150 km is unreal for F2-layer. Even under the deep minimum of solar activity in 2008-2009 daytime hmF2 according to ISR observations never was lower than 200-210 km. Therefore all RO hmF2 lower than 200 km should be considered as erroneous ones.

We accept the suggestion. For NmF2 and hmF2 comparison, we re-select co-located data pairs using altitude range (200,500), and re-calculate all the results; for EDP comparison, we use all the co-located data pairs.

(12) P6 L7 the over a year temporal segment Poor style

We modify the sentence to make it easy understand.

(13) P6 L12 Special attentions should be paid on the local time issue when CSES and COSMIC RO data are combined together. The phrase is not clear.

More information is added before this sentence, to make this sentence more clear and easy to understand.

(14)P6 L29 The RO Ne(h) profiles very often are not smooth at all. How such cases were developed?

We make some modification on this paragraph. The way to calculate the data at these selected altitudes is given in the following paragraph.

(15)P7 P13 also L13 as hollow circles pen circles

We accept the suggestion, and modify the paper as suggested.

(16)P8 L15 This is usual MRD (mean relative deviation) –there is no need to invent new definitions.

We accept the suggestion, and modify the paper. The equation is deleted.

(17)P9 Table 1 Hardly real accuracy of NmF2 determination requires 5 digits.

This is not an indication of precision, but to maintain the same number of significant digits as the power exponents are set to the same.

(18)P9 L27 There is another point to point out. Poor style.

We modify this sentence.

(19)P10 L6 CSES RO derived peak values are in very good agreement with COSMIC and ground based measurements. No comparisons with ground-based NmF2 observations are done in the paper.

This conclusion is based on the high consistency between COSMIC and CSES data, and on the COSMIC validation work conducted using ground-based ISR and ionosonde measurements, as discussed earlier in this passage. According to correlation transitive rule, we can get this conclusion.

We make some improvement here to make it easy to understand the logic behind the analysis.

(20)P10 L 28 As we know, the nighttime data has a more complex spatial distribution pattern compare to daytime data although daytime data are affected by solar radiation during day time.

We modify this sentence by adding more information, to make it clear.

(21)P10 L36 They suggested that COSMIC measurements are acceptable under geomagnetic disturbed conditions when comparing COSMIC RO data with observations from Sanya, a lower latitude ionosonde in China. Hardly one can agree with this statement. The equatorial anomaly introduces a spatial asymmetry especially during storm periods. This asymmetry should affect RO results.

COSMIC measurements are acceptable under geomagnetic disturbed conditions. This conclusion is from the reference paper by Hu et al. (2014) using data from 2008-2013 in Sanya Station, China. This conclusion may need more validation work. However, since it is published, we can reference its conclusion.

Anyway, we further modify this paragraph to make it more logical.

(22)Part 3.2 and Table 2 indicate excellent results RMSE <15 km This is a difference within the RO method obtained by two similar devices. But it should be stressed that the difference between RO hmF2 and real hmF2 may be different. The most accurate hmF2 provide ISR observations and only such comparisons may give a real estimate of RO hmF2 accuracy.

We agree with the referee's idea. In response, we modify the corresponding paragraph to make it logically arranged.

**Response to Referee 2**

First of all, we thank referee #2 for his/her advices and suggestions on our work. Response to the suggestions are as following.

(1) Line 30 The presismic character of observations from space is still under discussion. Please

avoid over-stretched statements about precursors that are still supported by poor statistical confirmation.

Response: As referee#2 points out, earthquake precursor (preseismic character) is still a debate. "Are there precursors helpful to earthquake prediction" is still a scientific problem. This paper is to validate the data obtained by the newly launched satellite. A general introduction to the satellite is necessary to give the readers complete information about the satellite. Therefore, a short introduction is given in this paragraph. The introduction is shortened from the CSES brochure. It is just an objective introduction, no mean to discuss problems with earthquake precursors.

(2) Page 6 Line 7 Please review the style.

We modify this sentence, and try to improve the English of this paper.

(3) Page 6 Line 12 Please clarify the definition of the time at which CSES and COSMIC data are compared.

The time criterion to select matching CSES and COSMIC event is defined in previous paragraph: the time difference between the matching occultation pairs is less than 30 min. Page 6 Line 12 stresses that COSMIC RO data covers all local time, while CSES RO data only covers 2 special local time due to its special orbit. Data users should pay attention to this.

In resonse to this suggestion, we revise this paragraph to give a more detail description.

(4) Page 9 Line 27 Change style.

We modify this sentence as suggested.

**Author's changes in manuscript**

All the changes are marked with red color and underline, which is provided in a separated file.

---

## Referee Report (RR1)

**Comments on the revised version of the paper**

The authors have introduced some recommended changes to the text.
However they state that RO observations provide reliable results which can be used both for physical analyses of the ionosphere and its monitoring and it is well-known and I told about this that the RO method has many limitations and not all Ne(h) RO profiles are reliable especially in the bottom- side.
Anyway they have demonstrated a good coincidence with COSMIC observations and that was the main purpose of the paper sufficient to be published.

The paper after minor corrections can be published in AG.
 **Some specific comments**

1. Line 26 Should be openly said that that bottom side Ne(h) profiles are less reliable than topside ones.
2. Line 14   on April
3. Line 9 There are different programs of IRS observations and slant observations  are also conducted when Ne(h) points belongs to different latitudes and longitudes.
4. Line 30   as much as possible
5. Line 28   0.7933% looks strange,  nobody uses more than 1-2 digits in %
6. Line 27   in a very good agreement with COSMIC and ground-based measurements
7. Line 31 not screening but manually scaled

---

## Author Response (AR2)

In response to the reviewer's suggestion, the following modifications are made.

1. Modification to the end of the abstract, as suggested by the reviewer: data at bottom side of the profiles are less reliable than that at the peak and topside regions.
2. Modification to the ISR observation. Since there are slant ISR observations, we limit our expression as: vertical ISR observations
3. Modification to the relative error expression, two digits are preserved for all the results in this paper.
4. Other small modifications are done according to the suggestions.